# Artificial Intelligence of Manufacturing Robotics Health Monitoring System by Semantic Modeling

**DOI:** 10.3390/mi13020300

**Published:** 2022-02-14

**Authors:** Han Sun, Yuan Yang, Jiachuan Yu, Zhisheng Zhang, Zhijie Xia, Jianxiong Zhu, Hui Zhang

**Affiliations:** School of Mechanical Engineering, Southeast University, Nanjing 211189, China; 230218029@seu.edu.cn (H.S.); yangyuancsi@163.com (Y.Y.); 230208026@seu.edu.cn (J.Y.); zhanghuihui@seu.edu.cn (H.Z.)

**Keywords:** robotic semantics, machine learning, piecewise regression, fault detection, fault diagnosis

## Abstract

Robotics is widely used in nearly all sorts of manufacturing. Steady performance and accurate movement of robotics are vital in quality control. Along with the coming of the Industry 4.0 era, oceans of sensor data from robotics are available, within which the health condition and faults are enclosed. Considering the growing complexity of the manufacturing system, an automatic and intelligent health-monitoring system is required to detect abnormalities of robotics in real-time to promote quality and reduce safety risks. Therefore, in this study, we designed a novel semantic-based modeling method for multistage robotic systems. Experiments show that sole modeling is not sufficient for multiple stages. We propose a descriptor to conclude the stages of robotic systems by learning from operational data. The descriptors are akin to a vocabulary of the systems; hence, semantic checking can be carried out to monitor the correctness of operations. Furthermore, the stage classification and its semantics were used to apply various regression models to each stage to monitor the quality of each operation. The proposed method was applied to a photovoltaic manufacturing system. Benchmarks on production datasets from actual factories show the effectiveness of the proposed method to realize an AI-enabled real-time health-monitoring system of robotics.

## 1. Introduction

Electric motors are an essential part of most manufacturing robotics. There are many kinds of motors that are capable of driving the system to accomplish cyclic, linear, or more compound actions. Usually, robotics is designed to do some dedicated tasks by controlling a group of motors with servo systems or computers. Without proper monitoring of the robotics, it may lead to the low quality of product, or even safety issues. Fortunately, modern servo motors usually come with various sensors, which help with the analysis of the whole system. The servo controller is responsible for precisely controling the speed and position of motors. It receives control signals and transforms them into a torque output to drive the motor. The mechanism of the servo controller involves a closed-loop system, which synchronously reads position and speed sensors of the motor to adjust the torque output continuously. Most servo motors would share sensor interfaces to the user for customized control. From the recorded sensor data, it is possible to manually check out whether the robotic system is following the right routine. However, when the complexity of the system grows, it is hard for humans to keep an eye on all motor signals, and an automatic verification method is required to replace human effort. Furthermore, even if the system is following the correct routine, it does not necessarily mean a healthy state. For instance, if there are foreign objects in the actuator, or the bearing is lacking lubricant oil, the quality of product may be affected; the duration of the system will be shortened too. In this study, we aim to utilize the sensor data of motors in a robotic system to verify the correctness of operations and detect anomalies.

Fault detection and diagnostics (FDD) has seen emerging demands in industrial intelligence systems. Various studies have been devoted to FDD for manufacturing, chemical engineering, commercial building, etc. [1,2,3,4]. In robotic systems, FDD is used for monitoring malfunctions and health status. It is essential to develop an effective model to describe the robotic system before FDD. In Reference [5], Wu et al. proposed a hierarchical functional model to disjoint a complex robotic system. The top-level system is divided as several major tasks, which are further resolved into multiple actions. However, the action definition is mostly manual; hence, there is a need for much labor when initializing a new system. Miyazawa et al. [6] offers a UML-based framework for the semantic modeling of robotics. The framework supports automated reasoning, but it still need manual work at initiating. In Reference [7], Zhao et al. takes the modeling further to recognize states and functions of a component automatically. The system they studied involves only discrete signals; the behaviors happen suddenly, so they may not be easily generalized to a random robotic system. In Reference [8], Y. Zhang et al. combines an expert system and signal processing into a knowledge transfer platform, which is capable of detecting faults with a set of basic knowledge-based rules to generalize to a variety of industrial systems. In Reference [9], Y. Zhang et al. detailed the work on gas turbine to characterize swirl with gray-box modeling. Zhang’s research provided a methodology base for our semantic modeling. Semantic analysis is widely used in natural language processing [10,11] and computer vision [12,13]. With proper modeling, it may be applicable to robotic FDD. The other research area of robotic FDD is health monitoring. Many works focus on a single component, such as remaining useful life prediction for a bearing [14,15,16,17] and anomaly detection [18,19]. However, on a multi-process multicomponent system, a single model may be inadequate. Piecewise regression is usually used for multistage process modeling. Conventional piecewise regression requires break point [20] or discontinuity estimation [21]. The semantic modeling could provide a more straightforward way for dataset partitioning.

The robotic equipment for PV manufacturing is composed of X-Y servomotors which are responsible for feeding in(out) and (un)loading the Plasma-Enhanced Chemical Vapor Deposition (PECVD) boat. The RCX motor moves horizontally, sending the boat into the reaction chamber, or taking it out. The travel range of RCX motor is around 0–3500 mm. The RCY motor moves vertically, loading and unloading the raw material or final product with the boat. The travel range of RCY motor is around 0–65 mm. Each motor reads current position (P), speed (V) and torque output (T) in real time. In the following text, we refer to the signals of RCX and RCY with a X-/Y- prefix respectively. An additional sensor is installed in the boat to read current load state, B; 0 means empty boat, and 1 means loaded boat. The boat state reflects the load of the motors; hence, it is correlation with the torque output of the motors. In this work, we propose a method to check if the system is operating in correct sequence and detect anomaly by building regression model for each semantic stage.

## 2. Proposed Method

### 2.1. Semantic Modeling of Robotic Systems

Our semantic system learns the model of a multistage robotic system by training on a few healthy cycle operations of the robotics. One full cycle contains all stage transmissions and the torque output of motors in healthy state usually follow the same pattern even though the duration of each individual operation may vary. The framework of the proposed system is shown in Figure 1. First, the system analyzes the individual signals of the system to segment the signals into several semantic states by machine learning techniques [13,22,23,24]. The states of all sensor signals combine into a full functional model of the system. The stage recognition is carried out in an automatic manner, in contrast to the work in Reference [13], which requires the manual definition of any operation stage of the system. This makes the method easily transferable to a different robotic system. The generalization capability will be verified in the experiment sections. The sequence of the stages reflects the semantics of the top-level function of the system. Considering the stages as an alphabet of the system, then the sequence of operations composes a “word”, on which spell checking could be applied to check the correctness of robotics “syntax”. However, correct operation does not guarantee healthy robotics; some vital signals, such as the vibrations and torque, could reflect the healthiness of the robotics. Therefore, a modeling strategy is needed to represent how the signals appear in healthy state. For each stage, a specific model that best describes its characteristics can be selected from a few candidate models. Besides that, the time after entering a stage is added as an input, which may be required to solve underfitting in several cases. One last feature of the proposed method to mention, there are cases when a stage may exhibit different phenomena with different circumstances in an operation sequence. The proposed method makes it possible to utilize former semantics as an additional feature for modeling. The semantic modeling is capable of learning and checking the sensor signals, and finally can be combined into a health monitoring system.

### 2.2. Robust Automatic Stage Learning

The multistage robotic system can be considered to be a piecewise function. To achieve the goal of semantic segmentation and fitting different function expression to each section, the section domain must be given. In other words, when a sample from the time series data is given, a domain must be determined first, and then the function of the domain is used for prediction of the sample. In the proposed method, a healthy operation cycle is used to learn the domain stages of the robotic. The number, sequence, and name of the operation stages are not required in advance and are concluded in an automatic manner. The original variables from the sensors can be divided into two categories. One class is the discrete variables, such as the B signal for boat state in our case. The other class is the continuous variables, such as P and V for current position and speed. The discrete variables are discriminative by themselves, while the continuous variables are not ideal to describe a certain state. The values of continuous variables do have aggregation features, but they cannot be easily determined by one clustering model. In addition, the direction of constantly changing states cannot be expressed by the original variable. In this section, we introduce a state descriptor to label both discrete and continuous variables from the sensors. Thereafter, the unique combinations of states are regarded as one stage of the robotic system without more prior knowledge.

#### 2.2.1. State Descriptor and Automatic Labeling

In Von Wright’s theory of action [5], there are four basic types of actions: happen, remain, disappear and remain absent. This theory thinks of a system as a series of instant state shifts. It focuses on the change of states, even a stable state is regarded as a shift from the state to itself. The pTp schema is used to illustrate the state changes. This theory is a potential describer for a complex robotic system. In our application of PV manufacturing robotics, the combination of states is more concerned rather than the transition of states. Two types of states are defined for each sensor variable, namely stays and changes. The change (C) state indicates that the variable is constantly changing; the stay (S) state indicates that the variable remains at a certain value. For discrete variables, there is only the S state, staying at a different value. It is denoted as S(*m*), where m is the event that it stays at. For continuous variables, the state can be S or C; S state is the same as discrete variables. The C state is denoted as C(*m*,*n*), where m is the original state, and *n* is the target state, meaning the variable is transiting from the *m* event to the *n* event.

With the state definition, the sensor variables can be represented by discrete states. However, labeling the states manually is very tedious work; thus, it is not feasible in most application situations, especially for complex robotic systems. In this work, we propose an automatic classification method suitable for the state definition mentioned above. There are several problems that must be noticed: Firstly, the variable in S state is not guaranteed to be constant, and it may introduce small noises. Secondly, the variable in C state may have a different slope; the classification model must be robust to the variance. We propose to first classify the time-series data into two categories: staying and changing. Then all the staying states are clustered into several centers, which are the m values with the S state. The C states are further labeled with the S states before and after a continuous C state. It is likely that a C state would transit from one S(*m*) state to the same S(*m*) state; it is still a value state.

The main difference of a C state and a S state is the gradient of the signal. The gradient of a S state is close to zero; a C state has a non-zero gradient. Given a specific time in the signal, it is not possible to judge the state from a sole value, so a window before this time is used. The window size, W, relies on the sampling rate and noise condition of the original signal. It must be robust to possible turbulence in the signal. A too-small W value will make the window easily affected by noises, while a too-large W value will be rigorous for classification. In our application, the P signals have very subtle noises, while V signals are subject to apparent noises. It is preferable to train a model that is suitable to any noise conditions, or a model that can be easily adjusted to transfer to a different condition. Empirically, W = 20 samples are selected, and they just a few times larger than maximum noise length for model training. With the window length, the signal is reshaped to a matrix of windows. It can be seen from Figure 2 that S state windows are nearly flat lines around a certain value, while C state windows are either line with a slope or a combination of above two types. For versatility, the window samples are first standardized to 0 mean, and then features can be extracted from S and C state samples for classification. One way is to use a neural network for direct wavelet classification; another way is to reduce the dimension of window samples by PCA, and classify from the principle features.

A sample signal from X_P signal on RC1 motor is used to train the classification model. Random noises are added to the training set according to the maximum noise range of all sensors to promote robustness. The metrics for evaluation is the precision of classification based on a predefined empirical labeling. The model is evaluated from many perspectives: Firstly, it is required that the model fits the training set well. Secondly, the model should generalize to the same sensor on the same motor. Thirdly, the model should generalize well on other sensors and other motors. Lastly, the model trained on RC1 should be reusable on RC2-6. Multilayer perceptron and PCA with SVM are compared for this task.

Figure 3 shows that MLP can achieve 99.94% accuracy on the training set. However, the model is not robust when used on a noisy signal. Figure 4 shows the result of using PCA and SVM. The staying signal windows shows apparent aggregation after dimension reduction to 2 principal components. An SVM model with radian base function kernel is easily trained with first two components. The result indicates that PCA + SVM has better generality for classifying noisy signals to staying and changing states.

#### 2.2.2. Learned Stages and Semantic Checking

Based on the states of each sensor variable, stages are defined as the combination of all states. Since the sensor signals are all discretized to states, the stage can be determined for each moment. The unique combinations of states are just the stages of the robotic system. In our application, we name a stage as a sequence of states in fixed order, (X_state, Y_state, Load_state). In the previous section, the signals can be labeled as S or C states. The arguments of the states are determined as such: First, the continuous S states are grouped, and the m value is determined by averaging the signal value in each group. Second, the continuous C states are determined by using the S state before and after the C group. For instance, when the system is feeding in a boat, X motor is moving from 0 position to 3500, Y motor is in zero position, the load state is zero and then the stage is described as (C(0,3500)-S(0)-S(0)). The full cycle of the system is shown in Figure 5, along with the proposed state descriptor. In Reference [5], Wu et al. used a rule template to check the correctness of operations. With the proposed stage combinations, the correctness of stage sequence can be verified by natural language processing (NLP) techniques. Consider the stage descriptor as an alphabet of the robotic system, and then the vocabulary of the system is all possible sequences of correct operations. In our application, it is supposed that a full action has 5–8 steps, so all correct “words” learned from healthy operations are recorded. For simplicity, the stage descriptors are mapped to the English alphabet, as shown in Figure 6d.

With the alphabet, a reference vocabulary can be generated. For instance, a full cycle of unloading the product from the reaction chamber is (a→b→c→d→e→f). The maximum word length for building the vocabulary depends on different scenarios. In our application, the vocabulary is all possible sequence of length 5. When used online, a recent sequence of 5 stages is extracted to form a current word. The word can be compared to predefined vocabulary to check the “spell”. There are plenty of methods for spell-checking; in this work, we demonstrate that semantic checking is possible by our stage descriptor by using a global edit distance technique. Edit distance, also known as Levenshtein distance, is a measure of the similarity between two strings. It calculates the number of deletions, insertions or substitutions required to transform from one string to another string. Therefore, to check the correctness of an operation sequence, we only need to calculate the Levenshtein distance of the sequence to correct sequences in the vocabulary, as shown in Figure 7. If a sequence is identical to any in the vocabulary, then the sequence is correct. Otherwise, the most likely sequence can also be found by selecting the word with the least editing distance. The incorrect stage and the context can be given for further diagnostics and fixing.

### 2.3. Piecewise Regression

Given enough sensor data, the torque output of each motor in a robotic system is usually deterministic. For single motor modeling, the regression model is a straightforward solution for torque prediction. If the online torque value does not conform to the regression model, most probability it could be an anomaly. However, the situation becomes much more complex with more motors and more operation states, so a single regression model may not be able to describe a compound process. The regression models use a specific formula with parameters to describe one process. Single regression model lacks flexibility when the processes have different formula or parameters in their nature. The linearity assumption is often broken when the state changes. Many works [25,26] introduced neural network to add non-linearity to the model, but it is still very hard to produce a general model suitable for various situations. In this section, we first show the shortcomings of using one regression model for the whole process, and then our proposed piecewise regression method based on semantic stage segmentation is presented. The proposed method trains a separate model for each stage. Besides, with the semantic segmentation, the previous semantic and elapsing time in current stage can be used as additional features for regression.

Table 1 shows the result of fitting multivariate linear models on RC1 motors. The overall error is not very high, but the models are still underfitting in some stages. After introducing non-linearity by MLP regressor, the average loss is reduced further, but the underfitting still exists as shown in Figure 8c,d. In practice, such a regression model will raise false alerts in the underfitted stages. Therefore, one single regression model is not adequate to describe a multistage system.

Previous studies of piecewise regression mostly segment the data by heuristic of discontinuity. Meanwhile, in our proposed method, the semantic stage presented in the above sections provides a very handy and accurate disjoint partitioning. Based on the semantic segmentation, the full dataset can be divided into groups according to the number of stages in the system. In each data group, an optimal regression model can be applied. The regression model is selected from a candidate model pool. In our application, we set linear model, 2-degree polynomial and MLP regressor as candidates. The optimal regression model is selected according to the prediction loss on the test set to prevent overfitting. The piecewise models exhibit better prediction loss than the single model, as shown in Figure 9.

## 3. Experiment and Evaluation

The data were collected from a PV manufacturing factory. The servo controllers of RCX and RCY read the current position (P), speed (V) and torque (T) of the motors. An additional binary pressure sensor is installed on the boat carried by the motors to monitor the state of load information, denoted as boat info (B). All sensors are synchronized with a sampling rate setting of 200 Hz. One manufacturing pipeline has six reaction chambers, scheduled by an additional Ladder robotics (LD). The whole system runs in ordinary production routine from 00:00 to 08:00. Each robotics for six chambers works through an average of 12 complete operation cycles. Various anomalies are artificially applied, such as foreign objects, incorrect controller command and lack of lubricant oil. The artificial anomalies are recorded as a reference to verify the effectiveness of the monitoring system.

One full cycle from reaction chamber number 2 is used to learn the stages and semantics of the system. Eleven stages are concluded from the training set, and a vocabulary of 26 words representing correct operation sequences is built. Figure 7b shows an incorrect operation detected by the model, and the closest operation sequence that it should follow is predicted.

Based on the stage segmentation result, the entire dataset is divided into 11 subsets, named a–k respective to stages. A regression model is trained on each subset with the first 1 or 2 cycles, depending on the length of dataset. Smaller subsets will need more cycle data to converge. As a result, a piecewise regression model is constructed by 11 sub-models. The piecewise regression model is used as such: for a test sample, a window of 20 samples before it is extracted to classify its state descriptors; then the stage it belongs to is determined; and, lastly, the sensor data are fed into the correspond piece model for the stage. The model can be used online for unhealthy operation detection. As comparison, the model is tested on the entire dataset along with single linear regression model and MLP regressor. In Table 2, it is clearly shown that, in most cases, the MLP regressor is better than linear regression. However, our proposed piecewise regression model is significantly superior to both, as the RMSE and MAE are much lower, while the R-square score is much higher. The result shows that the semantic modeling provides reliable segmentation for piecewise regression, and piecewise model is more suitable for a complex multistage robotic system.

## 4. Conclusions

In this paper, we presented a fault-detection system based on semantic modeling. A state descriptor was proposed to describe general motor signals, and it is robust to noises. The semantic stages of a robotic system were concluded with a few healthy running cycles as combinations of state descriptors. The automatically generated stages resemble a manual definition, while no prior knowledge of the system design is given. The semantic modeling method was verified on a photovoltaic manufacturing robotic system. The stages concluded from the modeling are almost the same as human cognition. We further applied semantic segmentation and analysis to detect malfunctions in operations. Levenshtein distance was used to detect faulty operations and to diagnose the most probable correct routine that it should be following. To monitor the quality of each robotic operation, piecewise regression was carried out to build a separate model for each semantic stage. It was shown that the regression performance is much better than training a single model on the entire dataset. The proposed system can be easily initialized on a new robotic system with only a few healthy operation cycles. The effectiveness was verified on a photovoltaic manufacturing robotics for fault detection. The generality is worth studying further on more complex and diverse types of robotic systems.

## Figures and Tables

**Figure 1 micromachines-13-00300-f001:**
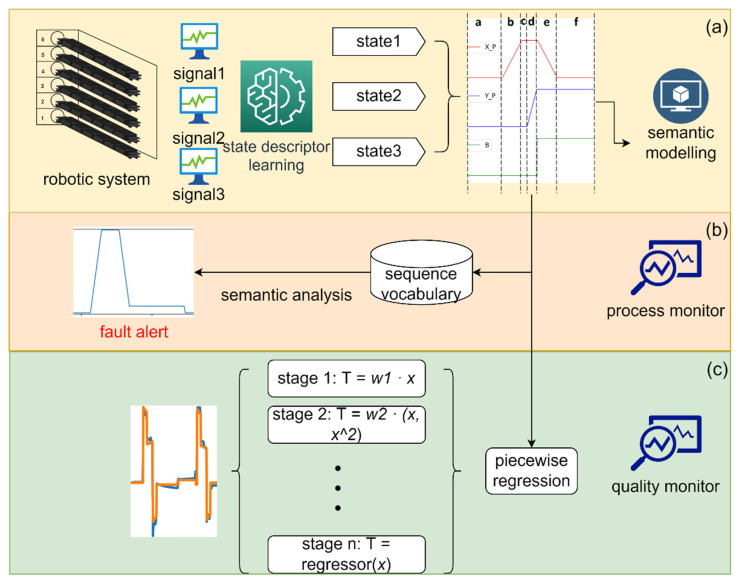
Framework of the proposed monitoring system. (**a**) Raw signals from the robotics are discretized and presented as state descriptors. The semantic model can be generated by state descriptors. (**b**) Semantic analysis can be carried out based on semantic modeling to find incorrect operations. (**c**) Time domain of signals is partitioned into segments. Piecewise regression is applied to each section to build health-monitoring model for each stage.

**Figure 2 micromachines-13-00300-f002:**
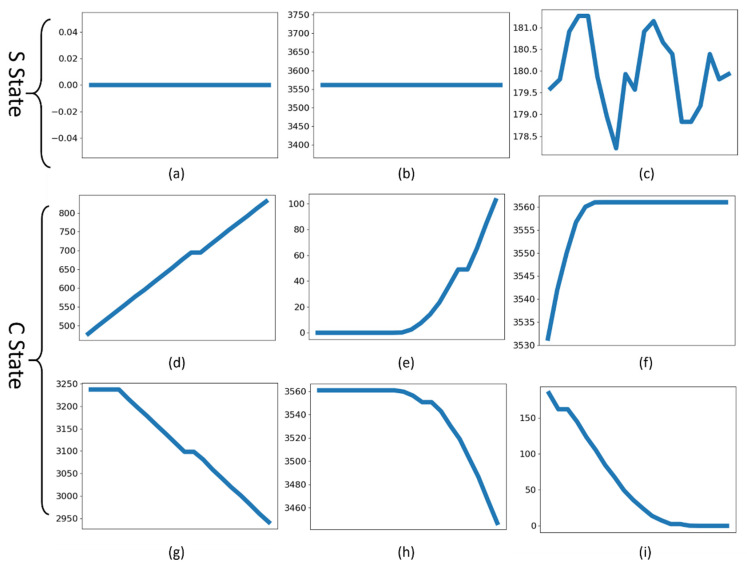
All possible shapes of signal window. (**a**) Staying at 0. (**b**) Staying at nonzero value. (**c**) Staying around 180 with noise. (**d**) Changing upward. (**e**,**h**) Staying to Chang. (**f**,**i**) Changing to staying. (**g**) Changing downward.

**Figure 3 micromachines-13-00300-f003:**
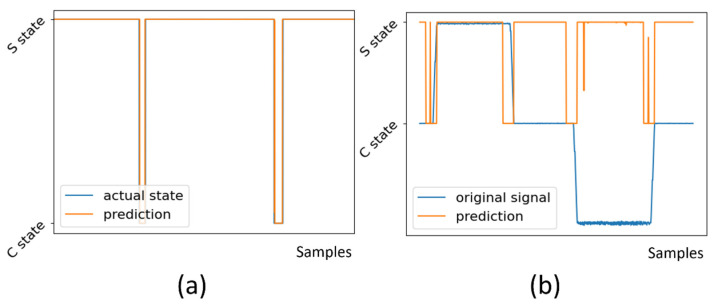
Prediction of S and C state by MLP. Value 1 means S state, and value 0 means C state: (**a**) prediction on signal with no noise, where the prediction accuracy is over 99.94%; (**b**) prediction on signal with noise. Wrong predictions occur as a result of noises.

**Figure 4 micromachines-13-00300-f004:**
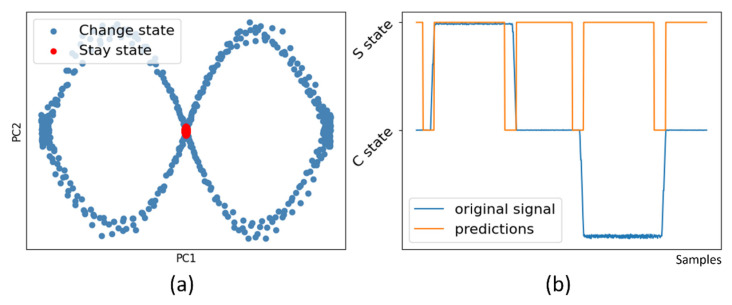
Prediction of S and C state by PCA + SVM. (**a**) First two principal components by PCA dimension reduction. Staying samples are aggregated in the center. Ascending and descending samples are aggregated in left and right ends. (**b**) Accurate prediction of state on a noisy signal.

**Figure 5 micromachines-13-00300-f005:**
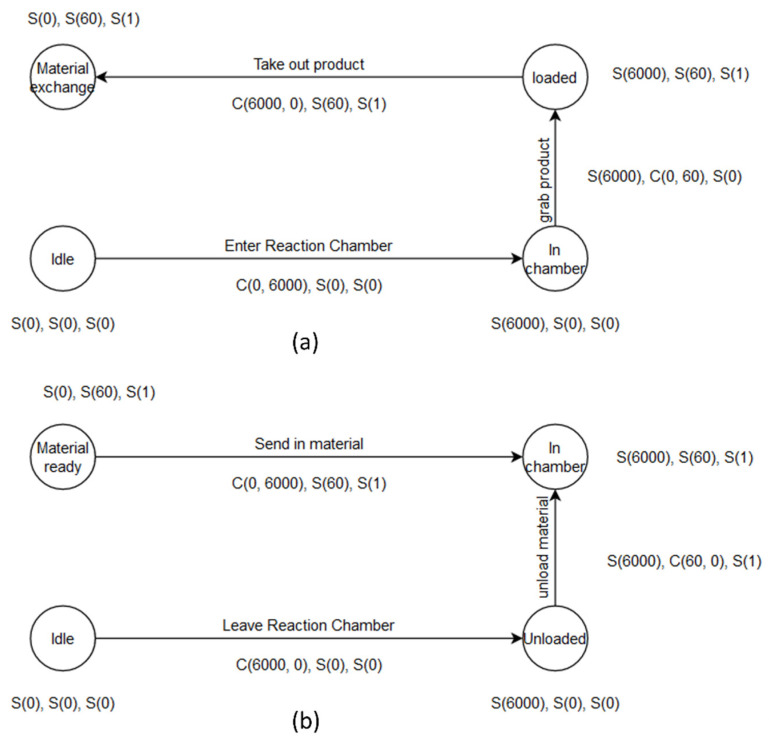
Full cycle of PV manufacturing robotic system described with proposed state descriptor: (**a**) product unload process and (**b**) raw-material load process.

**Figure 6 micromachines-13-00300-f006:**
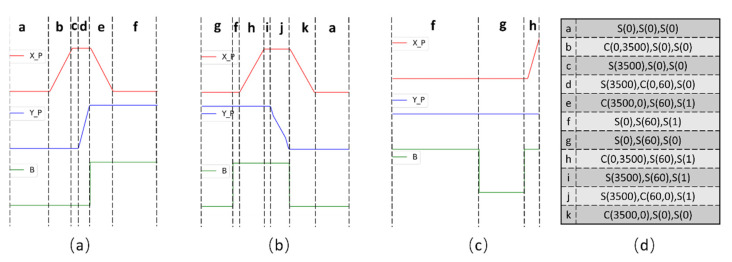
Main sematic sequences of the system: (**a**) unload product from reaction chamber, (**b**) load raw material into reaction chamber, (**c**) material exchange, and (**d**) reference table of simplified stage descriptor.

**Figure 7 micromachines-13-00300-f007:**
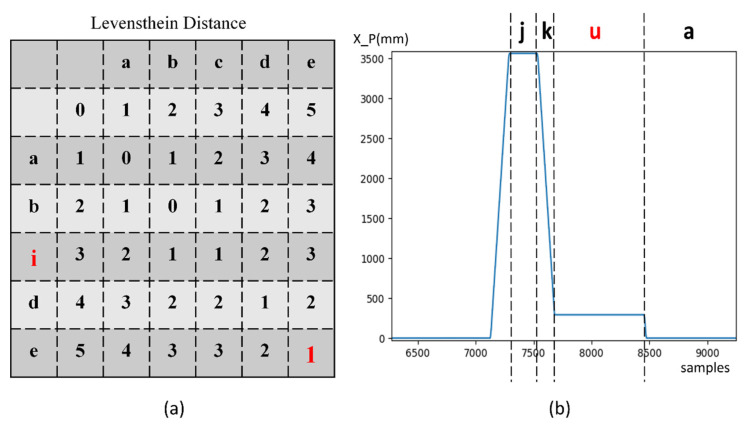
(**a**) Calculating Levenshtein distance of an incorrect operation to correct sequence in the vocabulary; ‘i’ in red color is an unexpected stage, the distance to ‘abcde’ is 1, (**b**) a detected fault operation, where “u” refers to an unknown stage. The closest sequence in vocabulary is {hijka} with edit distance 1.

**Figure 8 micromachines-13-00300-f008:**
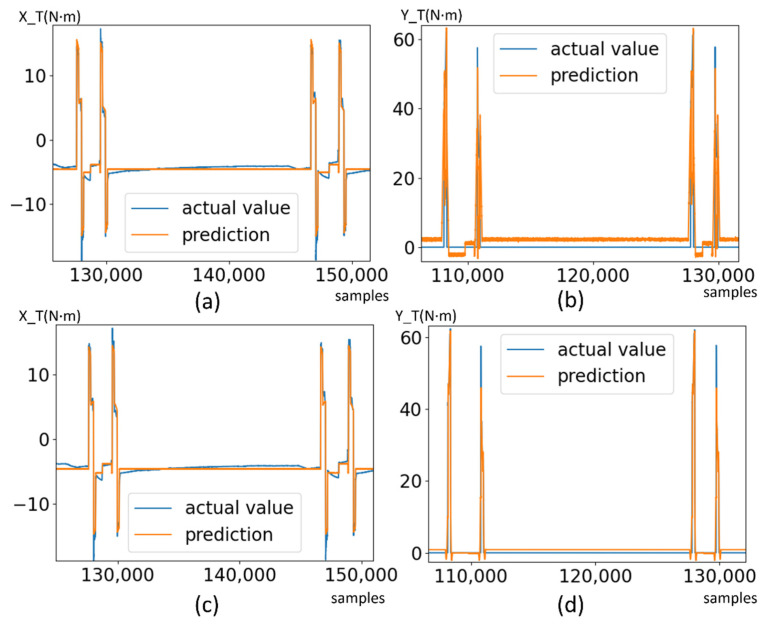
Fitting the dataset with a single regression model. (**a**,**b**) Prediction result of linear regression model on RC1 X/Y. (**c**,**d**) Prediction result of MLP regressor on RC1 X/Y. The result shows apparent underfitting in many stages.

**Figure 9 micromachines-13-00300-f009:**
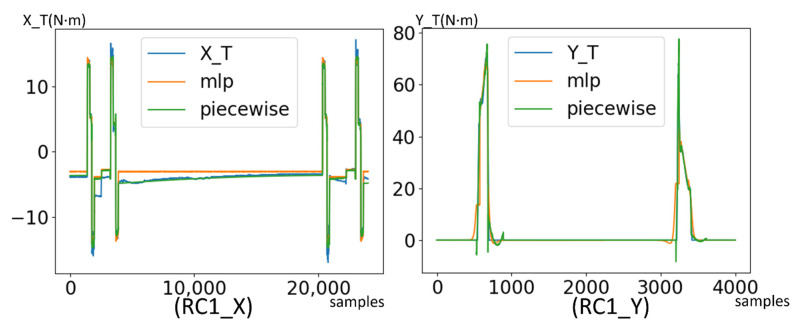
Comparison of regression performance between a single model and piecewise regression with semantic segmentation.

**Table 1 micromachines-13-00300-t001:** Evaluation metrics of linear regressor and MLP regressor trained on entire dataset.

Model	Motor	RMSE	$R^{2}$
Linear Regression	RC1_X	0.645	0.958
	RC1_Y	5.902	0.681
MLP Regression	RC1_X	0.590	0.965
	RC1_Y	2.024	0.962

**Table 2 micromachines-13-00300-t002:** Evaluation metrics show that the proposed method is superior to a single model in all 6 groups of motors.

Motor	MLP			LinearReg			Piecewise		
	RMSE	$R^{2}$	MAE	RMSE	$R^{2}$	MAE	RMSE	$R^{2}$	MAE
RC1_X	0.59	0.96	0.37	0.64	0.95	0.39	0.40	0.98	0.22
RC1_Y	1.86	0.91	0.89	5.77	0.12	3.08	0.62	0.99	0.07
RC2_X	1.21	0.84	1.02	0.79	0.93	0.59	0.55	0.96	0.37
RC2_Y	1.71	0.90	0.21	3.27	0.63	1.33	0.67	0.98	0.06
RC3_X	0.99	0.90	0.51	1.24	0.84	0.65	0.53	0.97	0.29
RC3_Y	2.10	0.92	0.38	3.21	0.14	1.11	0.96	0.98	0.31
RC4_X	0.61	0.92	0.38	0.69	0.90	0.43	0.48	0.95	0.27
RC4_Y	1.81	0.91	0.28	3.31	0.72	1.41	0.63	0.99	0.07
RC5_X	0.59	0.91	0.47	0.68	0.87	0.57	0.39	0.96	0.26
RC5_Y	1.33	0.91	0.12	2.45	0.71	0.79	0.49	0.98	0.04
RC6_X	0.77	0.90	0.59	0.82	0.89	0.64	0.49	0.96	0.33
RC6_Y	1.90	0.94	0.31	4.70	0.64	1.62	0.70	0.99	0.08

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
