# Peer review of "Artificial Intelligence of Manufacturing Robotics Health Monitoring System by Semantic Modeling"

_micromachines, 2022, doi:10.3390/mi13020300_

Round 1

Reviewer 1 Report

Abstract, line 11: “to defect abnoramlities” or “to detect abnormalities”?

English language should be revised, proof reading by native speaker, etc.

Line 71: “The travel range of RCX motor is around 0-3500”. Is there a measuring unit missing? Same question for line 73 and RCY motor with travel range 0-65.

Line 89: “Our semantic system learns the model of a multi-stage robotic system by training on a few healthy cycle operations of the robotics”. Do all healthy cycles generate same data? What is the justification for such small training data set (only few cycles)?

Line 102: Comma should go after word “Therefore”.

Figure 2. Axis labels are to small

Line 170: window is set to W=20 what? Seconds, operation repetitions, samples?

Figures 3 and 4, what do values given on x-axis represent, please add measuring unit and label

Typo on Figure 5: “Raction chamber” instead of “Reaction chamber”

Figures 7b and 8: x- and y-axis are missing measuring unit and label

Higher quality images should be used in paper

Figure 9: x- and y-axis are missing measuring unit and label

Author Response

Point 1: Typos in text and figure. Abstract, line 11: “to defect abnoramlities” or “to detect abnormalities”; Typo on Figure 5: “Raction chamber” instead of “Reaction chamber”;

Response 1: Typos in line 11, figure 5 and several other mistakes are fixed.

Point 2: Line 102: Comma should go after word “Therefore”.

Response 2: Punctuation are checked and revised, such as line 102.

Point 3: Units missing in text and figure. Line 71: “The travel range of RCX motor is around 0-3500”. Is there a measuring unit missing? Same question for line 73 and RCY motor with travel range 0-65; Figures 3 and 4, what do values given on x-axis represent, please add measuring unit and label; Figures 3 and 4, what do values given on x-axis represent, please add measuring unit and label; Figure 9: x- and y-axis are missing measuring unit and label

Response 3: Units are added in text(line 71&73) and in figures(4, 5, 7, 8, 9).

Point 4: Figure 2. Axis labels are to small; Higher quality images should be used in paper

Response 4:Axis labels are enlarged, irrelavant axis ticks are simplified. All figures are redrawn with higher resolution

Point 5: Line 170: window is set to W=20 what? Seconds, operation repetitions, samples?

Response 5: Line 170 the window is selected as 20 samples, unit is added.

Point 6: Line 89: “Our semantic system learns the model of a multi-stage robotic system by training on a few healthy cycle operations of the robotics”. Do all healthy cycles generate same data? What is the justification for such small training data set (only few cycles)?

Response 6: More explanation is added in line 89. Healthy cycles may vary in duration, but follows the same pattern for each stage. One cycle is enough for stage segmentation, the state descriptor is very robust, hence need not to retrain the classification model for new cases. The piecewise regression model may need more cycles to verify.In experiments, we used 2-3 cycles for training, 2 cycles for validation and the entire operation time for test.

Point 7: English language should be revised, proof reading by native speaker, etc.

Response 7: All authors have read through the manuscript again. Several necessary revisions are made. Proof reading by native speaker is also involved.

Reviewer 2 Report

This paper proposed a fault diagnosis method for multi-stage robotic systems, involving semantic modelling, semantic analysis for process monitoring and piecewise regression for quality monitoring. The paper needs some expansion with further justifications, explanations and comparisons/discussions. (1) Introduction should be improved to discuss on more relevant work re. the methodologies. The unsupervised learning stage is interesting – which involves in incorporating physics-guided/expert knowledge in the modelling process – this aspect should be emphasized, referring to “A knowledge transfer platform for fault diagnosis of industrial gas turbines” and “Grey-box modelling of the swirl characteristics in gas turbine combustion system”. (2) Figure 1 should be revised – check the texts and typos. (3) The comparison results in Sec. 2.3 seem a bit weak – more performance metrics, more regression methods, or more discussions could be included. Sec. 3 could be expanded to include more findings and discussions, or alternatively, it could become Sec. 2.4. Scientific merits could be further explained. (4) Again, in Conclusion, the “little human interference” aspect could be explained further, as this could be the main contribution of this paper.

Author Response

Point 1: Introduction should be improved to discuss on more relevant work re. the methodologies. The unsupervised learning stage is interesting – which involves in incorporating physics-guided/expert knowledge in the modelling process – this aspect should be emphasized, referring to “A knowledge transfer platform for fault diagnosis of industrial gas turbines” and “Grey-box modelling of the swirl characteristics in gas turbine combustion system”.

Response 1: We found the two papers you suggested very inspiring. Y. Zhang's work gives the methodology basis for our research. Relevant methodologies are emphasized in Introduction. After further consideration, we think it might not be suitable to call the method 'unsupervised', because we do used labels in training. The advantage of the method actually lies in its robustness and generality. In the experiments, it is also proved that the method is more robust than neural network models, since it can give more precise state classification on test data. Therefore, we have changed the explanations on stage learning to make it more proper.

Point 2: Figure 1 should be revised – check the texts and typos.

Response 2: Typos in text and figures are revised.

Point 3: The comparison results in Sec. 2.3 seem a bit weak – more performance metrics, more regression methods, or more discussions could be included. Sec. 3 could be expanded to include more findings and discussions, or alternatively, it could become Sec. 2.4. Scientific merits could be further explained. 

Response 3: We added one entire data set modelling(linear regression) for comparison. More performance metrics is added. Discussion on the results is complemented in Experiment and evaluation section.

Point 4: Again, in Conclusion, the “little human interference” aspect could be explained further, as this could be the main contribution of this paper.

Response 4: We made more justification in conclusion section. The most significant value is that the proposed state descriptor is universal in a variety of sensor signals. It is robust to noises, hence capable of classifying any new sensor signal into Change and Stay states without retraining in new situations. The only inputs for the proposed system for a new case is just a few healthy operation cycles. No prior knowledge of the robotics design or operating logic is required.

Round 2

Reviewer 2 Report

The comments have been addressed appropriately.